# Calculation of Three-dimensional Energy Product for Isotropic Nd$_2$Fe$_{14}$B Magnet

Namkyu Kim [1], Hee-Sung Han [2,3], Chul-Jin Choi [1], Ki-Suk Lee [3,*] and Jihoon Park [1,*]

[1] Powder Materials Division, Korea Institute of Materials Science, Changwon 51508, Korea
[2] Center for X-ray Optics, Lawrence Berkeley National Laboratory, Berkeley, CA 94720, USA
[3] Department of Materials Science and Engineering, Ulsan National Institute of Science and Technology (UNIST), Ulsan 44919, Korea
* Correspondence: kisuk@unist.ac.kr (K.-S.L.); jpark@kims.re.kr (J.P.);
Tel.: +82-52-217-2340 (K.-S.L.); +82-55-280-3245 (J.P.)

**Abstract:** A conventional energy product calculated by the product of the *B*-field and the *H*-field is not sufficient for representing the performance of a magnet because it considers the homogeneous and only the uniaxial magnetic properties of the magnet. The conventional energy product has been compared with another energy product obtained by integrating the scalar product of the *B*-field and the *H*-field of each cell composed of the three-dimensional components. We investigated a model system by micromagnetic simulation using finite differential method (FDM) and calculated the full hysteresis of the magnet. The model system of a Nd$_2$Fe$_{14}$B magnet composed of grains with a diameter of about 100 nm was assumed. In the case of the isotropic multi-grain magnet, the energy product calculated by the integration method was 28% larger than the energy product obtained by the conventional way, although a discrepancy between the distribution of the magnetizations and the demagnetizing fields at the reversal process resulted in the decrease of the energy product.

**Keywords:** permanent magnet; energy product; micromagnetic simulation; multidomain reversal

## 1. Introduction

With the explosive rise of interest in sustainable energy production and eco-friendly transportation, including electric vehicles, the importance of energy conversion between electricity and kinetic energy has increased. Because a high-performance permanent magnet plays a key role in the efficiency of energy conversion through generators and motors, a number of permanent magnets have been investigated [1–3]. In order to be used in generators and motors, it is important to evaluate how much magnetic energy is emitted by permanent magnets.

Magnetic energy is represented by the energy product *BH*, which is twice the energy stored in the stray field outside the magnet. It can be obtained from the volume integral of the square of the stray field outside the magnet or from the volume integral of the dot product between the demagnetizing field $H_d$ and the internal magnetic flux density *B*, whose relation is shown in Equation (1) below. It is not only hard to measure the magnetic field distributed in the free space, but also the magnetic field on a closed surface. For this reason, the energy product can be used for a figure of merit to estimate the performance of the permanent magnet, whose values are proportional to the energy from the stray field outside the magnet [4].

$$E_{\text{out}} = \frac{1}{2}E_{in} = \frac{1}{2}\int_a \mu_0 H^2 \mathrm{d}V = -\frac{1}{2}\int_i B \cdot H \mathrm{d}V. \tag{1}$$

The energy product is usually calculated as the measured magnetization in the direction of the applied magnetic field. The energy product is calculated from the dot product

between the magnetic flux density $B$ and the magnetic field $H$, and the maximum energy product ($BH_{\max}$) is widely used as a figure of merit for evaluating the performance of hard magnetic materials. However, there are some missing points for the usage of the $BH_{\max}$. Firstly, $BH_{\max}$ is the highest value of the energy products among various magnitudes of the magnetic field. The energy products obtained from the magnetic hysteresis of a magnet loop can supply the expectation value of the energy product of magnets for different shapes, although it does not perfectly match the energy product of the magnets [5,6]. The only meaningful value for a system is an energy product at the remanent state, which is determined from the demagnetizing field by the shape of the magnet. Another missing point comes from how the energy product is calculated. Although the energy product is usually given as the area of a rectangle in a *B-H* hysteresis loop, there is a clear difference between the energy product and the real value of the energy product calculated from its original physical meaning. One difference is that the conventional energy product assumes the magnetization and the demagnetizing field as uniaxial components in spite of the three-dimensional emission of the magnetic field. Another difference comes from the discrepancy between the distribution of the magnetizations and the demagnetizing fields in the magnet. Because the permanent magnets are not used in the saturation state, the influence by the internal magnetic state during magnetic switching is inevitable for determining the energy product of the magnet. Most studies on the switching process cover the dynamic reversal process of soft magnets [7–12], and there are some research works that investigate the reversal mechanism of permanent magnets [13–15]. The reversal process is studied in order to estimate the effect of temperature [15], exchange spring [13], and grain boundary diffusion [14]. However, the effect of the reversal mode on the energy product has scarcely been studied despite its importance in estimating the practical magnetic energy of the permanent magnets. One of the reasons why there is a lack of studies devoted to estimating the energy product based on the theory is the difficulty in measuring the internal state of the magnet. Finally, an easy way to estimate the energy product, which multiplies the average values of $B$ and $H$, is widely used. Recent advances in computing technology have made it possible to investigate the complex magnetic properties inside magnets.

In this paper, we investigated a three-dimensionally integrated energy product from the magnetization and demagnetizing fields of cells to obtain the correct energy product; this was then compared to the energy product calculated using the conventional method.

## 2. Materials and Methods

We adopted an $Nd_2Fe_{14}B$ multi-grain magnet with dimensions of $2000 \times 2000 \times 1000 \, nm^3$, as illustrated in Figure 1. The material is commonly used with high coercive force and the energy product, and it has been investigated with the use of various models [14–19]. In order to investigate the magnetic properties during multidomain reversal, we assumed that a model magnet consists of grains with a diameter of 100 nm, and that the uniaxial magnetic anisotropy direction of the grains is randomly distributed. For the micromagnetic simulation, the unidirectional magnetic anisotropy constant of $4.9 \times 10^6 \, J/m^3$ was used. The saturation polarization was assumed to be $1.26 \, MA/m^3$, and we set the exchange stiffness constant to $3.0 \times 10^{-12} \, J/m$ [20].

To investigate the model system, a full hysteresis loop was simulated with Mumax3 [21,22], which implements simulations in magnetic solids described by a finite differential micromagnetic solver. In the finite differential calculation of the micromagnetic properties, a cubic cell of $10 \times 10 \times 10 \, nm^3$ was assumed, which was larger than the exchange length of the magnetic material. While the larger cell size can make a difference with the use of correct magnetic quantities of the materials, it is sufficient to investigate the tendency of the magnetizations and the demagnetizing fields during the switching process [23,24]. The hysteresis loop was computed through the minimization of the Gibbs free energy for an external magnetic field ranging from $-10$ to $+10$ T, which was ap-

plied along the cartesian *z*-axis. The total magnetic Gibbs free energy is described by the following equation:

$$E = E_{ex} + E_K + E_d + E_{Zeeman}, \tag{2}$$

where $E_{ex}$, $E_K$, $E_d$, and $E_{Zeeman}$ are the exchange energy, the anisotropy energy, the magnetostatic energy, and the Zeeman energy, respectively.

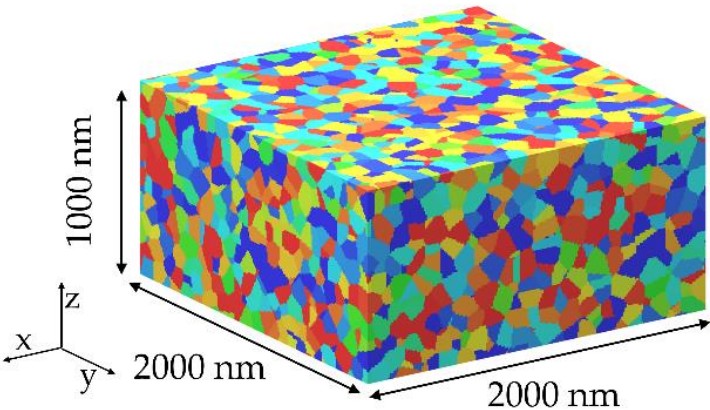

**Figure 1.** Schematic image of the $Nd_2Fe_{14}B$ magnet consisting of 4871 grains with randomly distributed anisotropy directions.

### 3. Simulation Results

The calculated magnetic hysteresis loops are represented in Figure 2a. The black solid line follows the magnetization ($\mu_0 M$) with respect to the applied magnetic field ($H_{app}$) from the simulation results. One of the main points is the difference between the methods used to obtain the values of the energy product ($BH$), which are expressed in Figure 2b. The orange line and squares indicate the energy products calculated in the conventional way, in which the values are from the product between the *B*-field and the *H*-field. The *B*-field is the sum of the magnetization along the direction of the applied magnetic field (+*z* in this simulation), while the *H*-field consists of both the applied field and the demagnetizing field along the +*z* direction. To reflect the experimentally measured values, the average values for the system were used. The violet line with triangle symbols represents the energy product calculated by integrating the energy product of each cell, which considers only a one-dimensional component along the magnetic field direction. On the other hand, the olive line and circle symbols in Figure 2b are the energy products calculated by the integration of the energy products of each cell; these energy products were obtained from the inner product between the *B*-field and the *H*-field. The three-dimensional energy product of each cell was calculated by

$$B \cdot H = B_x H_x + B_y H_y + B_z H_z, \tag{3}$$

where the *B*-field and the *H*-field follow the indicated direction.

There are definite gaps between the energy product from the conventional method and the integration method, which can be explained by the simple mathematical relation expressed below.

$$\int_i B \cdot H dV \ \neq \ \overline{B} \cdot \overline{H}. \tag{4}$$

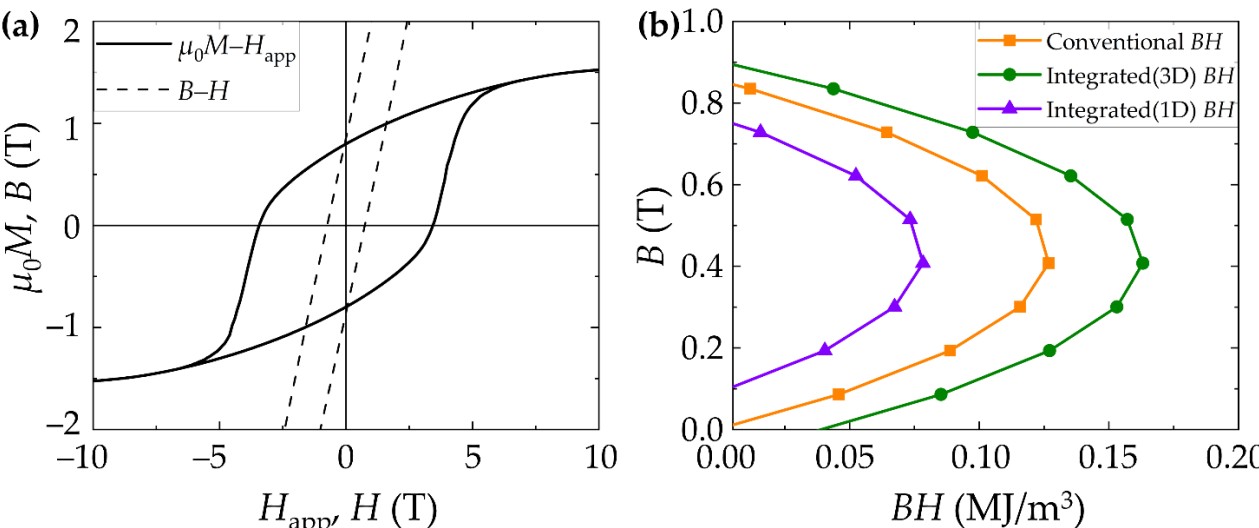

**Figure 2.** (**a**) Hysteresis loops of magnetization $\mu_0 M$ with the applied magnetic field $H_{app}$, and the magnetic flux density $B$ versus the magnetic field strength $H$ (=$H_{app}$ + $H_d$). (**b**) Energy product $BH$ calculated in three different ways.

The magnetizations and the demagnetizing fields during the reversal process were also investigated. Figure 3a represents the magnetization of each cell in color, while Figure 3b shows that of the demagnetizing field. It must be noted that the color indicates the magnitude along the +$z$ direction. With the magnetic field of 10 T, all the cells were saturated along the +$z$ direction, which is represented by the image filled in with the color red. With the reduction of the applied magnetic field, the magnitude of the magnetization was reduced and changed into the color blue. At the remanent state without the application of the magnetic field, the magnetizations were distributed to certain values on a grain-by-grain basis, which still have a positive value averaging to remanent magnetization ($M_r$). What we are focusing on is the difference of tendency between the magnetization and the demagnetizing field, and it can be easily understood from the close-up image of a cube envelopted in dashed lines. While the magnetizations in the same grain have similar values due to exchange interaction, the magnitudes of the demagnetizing fields are differently distributed in a grain. The edge of the grain has stronger magnitudes of the demagnetizing fields.

To clearly compare this difference, the histograms for cells with certain values of magnetizations and demagnetizing fields were investigated and represented in Figure 4. At the saturation state, the values of the magnetizations are distributed across a narrow range with high intensity, whereas the values of the demagnetizing fields follow a different distribution from that of the magnetization. The magnetizations were determined by the combination of the easy axis direction of the grain and the strength of the applied magnetic field, while the demagnetizing fields are associated with the magnetizations and the geometry of the whole system. Due to the exchange coupling, the magnetizations (Figure 4a) have almost the same value within the same grain, but they have different values depending on the grain because the anisotropy direction of each grain is different. On the other hand, the demagnetizing fields (Figure 4b) follow a distribution similar to the Gaussian distribution regardless of the magnitude of the applied magnetic field [25]. This discrepancy between the magnetizations and the demagnetizing fields results in the gap between the energy products calculated from the average values of the *B*-field and the *H*-field and from the integral of each cell. The distribution of the demagnetizing fields is speculated by cumulative distribution function (CDF) in Figure 4c. Empirical CDF from the simulation is plotted as a purple line, and for comparison, the standard normal CDF is plotted as black scatter. The empirical CDF almost coincides with the standard normal

CDF. Therefore, it is justified to approximate the demagnetizing field distribution with a Gaussian.

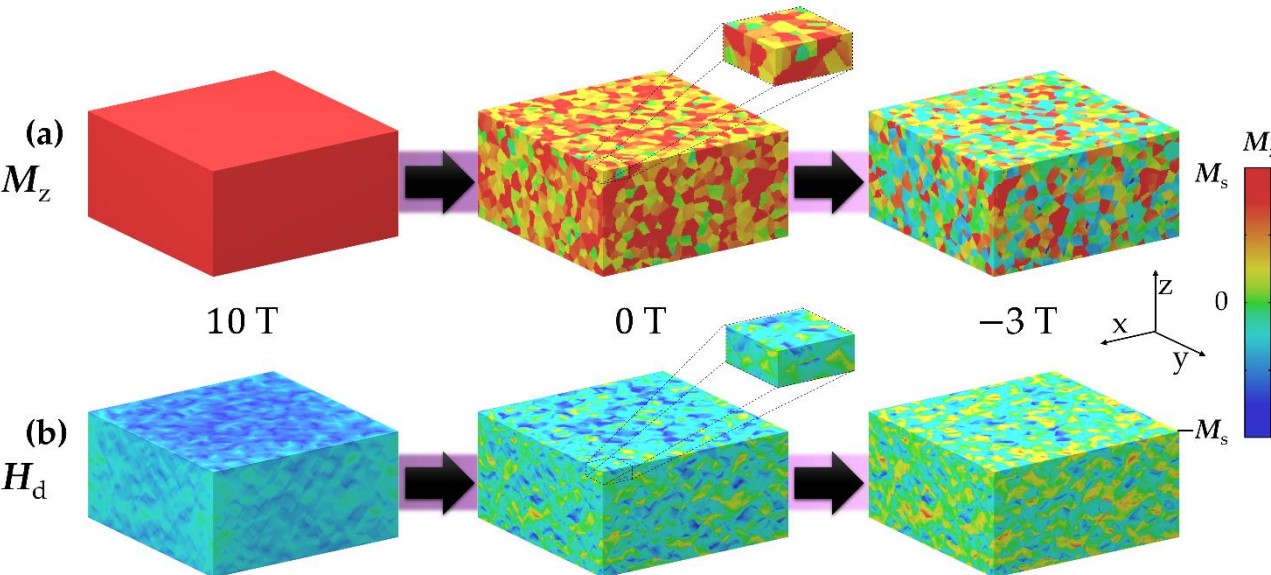

**Figure 3.** (**a**) Magnetizations and (**b**) demagnetizing fields of the model magnet at 10 T (saturation), 0 T (remanent state), −3 T (during reversal). Red and blue colors correspond to $M_s$ and $−M_s$, respectively.

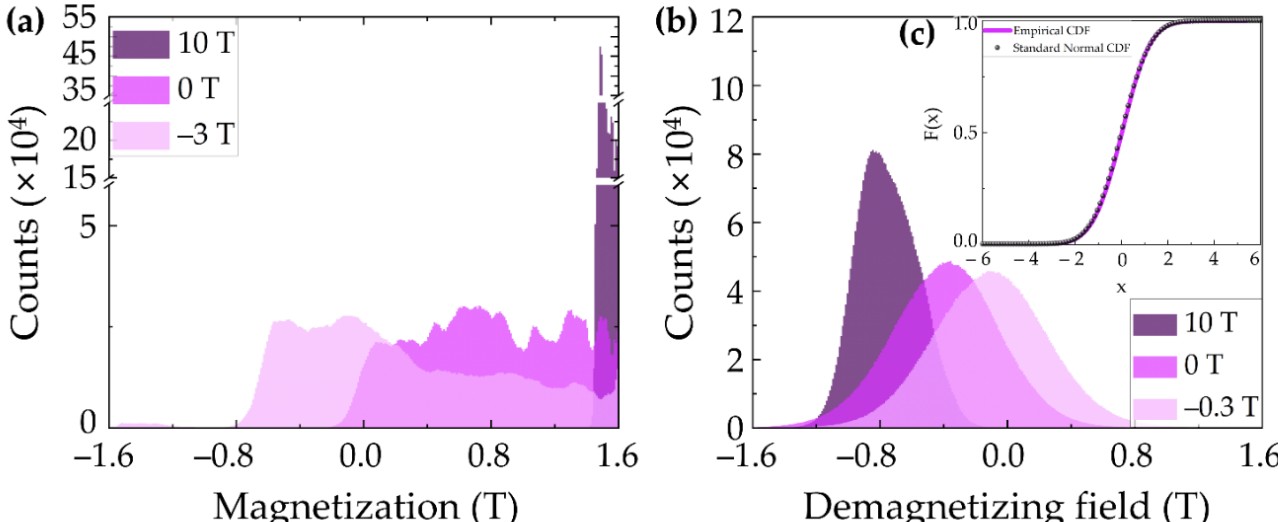

**Figure 4.** Histograms for cells with certain values for (**a**) magnetizations and (**b**) demagnetizing fields at 10 T (saturation), 0 T (remanent state), and −3 T (during reversal). (**c**) Cumulative distribution function (CDF) for the computed demagnetizing field at the remanent state, and the standard normal CDF by the mean and the standard deviation of the demagnetizing field in each cell.

The demagnetizing factor ($N = −H_d/M_z$) is a geometry-dependent value representing the magnet, which refers to the proportionality of the demagnetizing field with the magnetization [26]. To clearly show the mismatch between the magnetization and the demagnetizing field, the demagnetizing factor of each cell was investigated; a histogram for the distribution of the demagnetizing factor is given in Figure 5. In the saturation state, the demagnetizing factor is affected by the distribution of the demagnetizing field and follows a narrow Gaussian distribution because the magnetizations in the magnet are aligned along the applied field direction. On the other hand, in the case of the remanent

state, the demagnetizing factor spreads over a wide range due to the mismatch between the magnetizations and the demagnetizing fields.

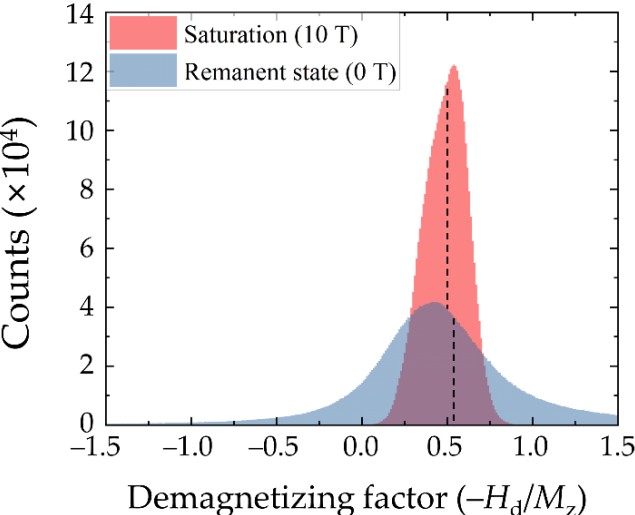

**Figure 5.** Distribution of the demagnetization factors ($-H_d/M_z$) of each element at saturation (red) and in the remanent state (blue). The black dashed lines indicate the 10% truncated means.

For these differences in distribution between the magnetization and the demagnetizing field during the magnetic reversal, the energy product calculated from the integration of each element is inevitably different from the energy product given by the product of the mean *B* and the mean *H*.

## 4. Conclusions

In summary, we have explored magnetization and the demagnetizing field via micromagnetic simulations to clarify the energy product in a multidomain reversal mode. More specifically, the energy product calculated by the product of the mean value of the B-field and the H-field was compared with the energy product calculated faithfully to their essence while considering the three-dimensional components of magnetization and the demagnetizing field. The simulation results show the obvious degradation of the energy product by comparing the one-dimensional energy product calculated by integration because the distributions of the magnetizations and the demagnetizing fields are different in the multidomain reversal mode. However, the energy product considering three-dimensional components is 28% larger than the energy product calculated in the conventional way. Therefore, to obtain the correct energy product, the magnetic alignment, the demagnetizing factor determined by the shape of the magnet, and the reversal mode should be considered.

**Author Contributions:** Conceptualization, N.K.; methodology, N.K. and H.-S.H.; software, H.-S.H.; validation, J.P. and K.-S.L.; formal analysis, N.K.; investigation, N.K.; resources, N.K.; data curation, N.K.; writing—original draft preparation, N.K.; writing—review and editing, K.-S.L. and J.P.; visualization, N.K.; supervision, C.-J.C.; project administration, J.P.; funding acquisition, C.-J.C. All authors have read and agreed to the published version of the manuscript.

**Funding:** This work was supported by Future Materials Discovery Program through the National Research Foundation of Korea (NRF) funded by the Ministry of Science and ICT (2016M3D1A1027835) and by the Korea Institute of Energy Technology Evaluation and Planning (KETEP) grant funded by the Ministry of Trade, Industry and Energy (MOTIE) of Korea (20192010106850, Development of magnetic materials for IE4 class motor).

**Institutional Review Board Statement:** Not applicable.

**Informed Consent Statement:** Not applicable.

**Data Availability Statement:** Not applicable.

**Conflicts of Interest:** The authors declare that they have no known competing financial interests or personal relationships that could have appeared to influence the work reported in this paper.

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
