# Peer review of "Calculation of Three-dimensional Energy Product for Isotropic Nd2Fe14B Magnet"

_applsci, doi:10.3390/app12157887_

Round 1

Reviewer 1 Report

The authors of the work " Degradation of Energy Product by Multidomain Reversal Mode" analysed the model system by means of micromagnetic simulation using the finite difference method and calculated a loop of magnetic hysteresis. A Nd2Fe14B magnet composed of grains with a diameter of about 100 nm was used for the modelling.

Nevertheless, I have some important remarks.

The work would be more valuable if the authors compared their results with the experimental results of other authors.

There are many works in the literature on the modeled material (Nd ..), unfortunately the authors do not cite important works describing the magnetic properties of the material for example (1.D.Givord, H.S.Li, J.M.Moreau, Magnetic properties and crystal structure of Nd2Fe14B Solid State Communications,Volume 50, Issue 6, 1984, Pages 497-499, https://doi.org/10.1016/0038-1098(84)90315-6.

2. Ramlan et al 2016 Crystal structure and magnetic properties of Nd2Fe14B powder prepared by using high energy milling from elements metal Nd,Fe,B powders J. Phys.: Conf. Ser. 776 012013 https://doi.org/10.1088/1742-6596/776/1/012013

3. L. Zhang et al., Journal of Alloys and Compounds 461 (2008) 351-354, doi:10.1016/j.jallcom.2007.06.093

4. T. Iida, T. Saito, K. Shinagawa and T. Tsushima, Field-induced spin reorientation in Nd2Fe14B and Er2Fel4 B, Journal of Magnetism and Magnetic Materials 104-107 (1992) 1363-1364)

 The authors should analyse the results of the modelling with the magnetic properties of Nd2Fe14B available in the wide literature.

Additionally, DOI numbers should be added to the reference list if assigned.

Author Response

Comment #1

The work would be more valuable if the authors compared their results with the experimental results of other authors.

There are many works in the literature on the modeled material (Nd ..), unfortunately the authors do not cite important works describing the magnetic properties of the material for example(

1.D.Givord, H.S.Li, J.M.Moreau, Magnetic properties and crystal structure of Nd2Fe14B Solid State Communications,Volume 50, Issue 6, 1984, Pages 497-499, https://doi.org/10.1016/0038-1098(84)90315-6.

  1. Ramlan et al 2016 Crystal structure and magnetic properties of Nd2Fe14B powder prepared by using high energy milling from elements metal Nd,Fe,B powders J. Phys.: Conf. Ser. 776 012013 https://doi.org/10.1088/1742-6596/776/1/012013
  2. L. Zhang et al., Journal of Alloys and Compounds 461 (2008) 351-354, doi:10.1016/j.jallcom.2007.06.093
  3. T. Iida, T. Saito, K. Shinagawa and T. Tsushima, Field-induced spin reorientation in Nd2Fe14B and Er2Fel4 B, Journal of Magnetism and Magnetic Materials 104-107 (1992) 1363-1364)

The authors should analyse the results of the modelling with the magnetic properties of Nd2Fe14B available in the wide literature.

Response #1

We are sorry about missing important works concerning with Nd2Fe14B magnet. To enhance the motivation for studying the effect of multidomain reversal mode based on Nd2Fe14B magnet, some explanation about using Nd magnet and references are reinforced.

(chapter 2, paragraph 1) There The material is commonly used with high coercive force and the energy product, and it has been investigated with various model [14-19].

Additional References

  1. Wu, D.; Yue, M.; Liu, W.Q.; Chen, J.W.; Yi, X.F. Magnetic domain switching in Nd-Fe-B sintered magnets with superior magnetic properties. Materials Research Letters 2018, 6, 255-260, doi:10.1080/21663831.2018.1437571.
  2. Nishino, M.; Uysal, I.E.; Hinokihara, T.; Miyashita, S. Dynamical aspects of magnetization reversal in the neodymium permanent magnet by a stochastic Landau-Lifshitz-Gilbert simulation at finite temperature: Real-time dynamics and quantitative estimation of coercive force. Physical Review B 2020, 102, doi:10.1103/PhysRevB.102.020413.
  3. Givord, D.; Li, H.S.; Moreau, J.M. Magnetic properties and crystal structure of Nd2Fe14B. Solid State Communications 1984, 50, 497-499, doi:10.1016/0038-1098(84)90315-6.
  4. Iida, T.; Saito, T.; Shinagawa, K.; Tsushima, T. Field-induced spin reorientation in Nd2Fe14B and Er2Fe14B. Journal of Magnetism and Magnetic Materials 1992, 104-107, 1363-1364, doi:10.1016/0304-8853(92)90620-4.
  5. Mo, W.; Zhang, L.; Shan, A.; Cao, L.; Wu, J.; Komuro, M. Improvement of magnetic properties and corrosion resistance of NdFeB magnets by intergranular addition of MgO. Journal of Alloys and Compounds 2008, 461, 351-354, doi:10.1016/j.jallcom.2007.06.093.
  6. Ramlan; Muljadi; Sardjono, P.; Gulo, F.; Setiabudidaya, D. Crystal structure and magnetic properties of Nd2Fe14B powder prepared by using high energy milling from elements metal Nd,Fe,B powders. Journal of Physics: Conference Series 2016, 776, doi:10.1088/1742-6596/776/1/012013.

Comment #2

Additionally, DOI numbers should be added to the reference list if assigned.

Response #2

We added DOI numbers for the all references. 

Reviewer 2 Report

The paper describes the measurements of NdFeB multigrain magnets.

The introduction section starts with the description of the motivation. This description is very general, and not totally true, because there are many research started to focus on the development of reluctance machines.Iit should be more focused on the given problem

The following sentences should be expanded and explained better: " The switching process is already investigated by many researchers ", it would be important to discuss the missing points and the advances of similar research works to show the state of the art and the importance of the topic.

It would be important to show the novelty of the paper in comparison with the selected, similar methods from the literature. This part is missing from the introduction.

The Simulation Results part of the paper describes the measurement, but it is not clear, that the given results are how new for the results of other papers.

Author Response

Response to reviewer #2:

Response

We are thankful for the referee’s statement about our manuscript, and we were pleased to revise it with your suggestion.

Comment #1

The paper describes the measurements of NdFeB multigrain magnets.

The introduction section starts with the description of the motivation. This description is very general, and not totally true, because there are many research started to focus on the development of reluctance machines. It should be more focused on the given problem.

The following sentences should be expanded and explained better: " The switching process is already investigated by many researchers ", it would be important to discuss the missing points and the advances of similar research works to show the state of the art and the importance of the topic.

It would be important to show the novelty of the paper in comparison with the selected, similar methods from the literature. This part is missing from the introduction.

Response #1

As the referee suggested, the introduction part has been rewritten to clarify the novelty of our work with relevant references. Please see the 3-5 paragraph in our revised manuscript and see Refs. [11-14].

The difference comes from distribution of the magnetizations and the demagnetizing fields in the magnet. Thus, the influence by internal magnetic state during magnetic switching is inevitable for the energy product. The switching process is already investigated by many researchers, [7-10] however, its effect on energy product of the permanent magnets has not been studied.

Therefore, in this paper, we investigated integrated energy product from magnetization and demagnetizing fields of cells to obtain the correct energy product, and compared it to the energy product calculated by the conventional method.

Firstly, the conventional energy product assumes the magnetization and the demagnetizing field as uniaxial components in spite of the three-dimensional emission of the magnetic field. Another difference comes from the discrepancy between the distribution of the magnetizations and the demagnetizing fields in the magnet. Because the permanent magnets are not used in saturation state, the influence by the internal magnetic state during magnetic switching is inevitable for determining the energy product of the magnet. Most of studies for switching process cover the dynamic reversal process of the soft mag-nets [[7-12], and there are some researches for investigating reversal mechanism of permanent magnets [13-15]. The reversal process is studied for estimating the effect of temperature [15], exchange-spring [13], and grain boundary diffusion [14].

However, the effect of the reversal mode on the energy product has scarcely been studied despite its importance in estimating the practical magnetic energy of the permanent magnets. At first, the reason why the lack of studies for estimating the energy product based on the theory is the difficulty in measuring the internal state of the magnet. Last but not least, the easy way to estimate the energy product, which multiplies average values of B and H, is widely used. Recently, with the development of the computing technology, studies for magnetic proper-ties inside the magnet by computational methods became further easier.

Therefore, in this paper, we investigated three-dimensionally integrated energy product from magnetization and demagnetizing fields of cells to obtain the correct energy product, and compared it to the energy product calculated by the conventional method.

Additional References

  1. Leighton, B.; Pereira, A.; Escrig, J. Reversal modes in asymmetric Ni nanowires. Journal of Magnetism and Magnetic Materials 2012, 324, 3829-3833, doi:10.1016/j.jmmm.2012.06.023.
  2. Skomski, R.; Ke, L.; Kramer, M.J.; Anderson, I.E.; Wang, C.Z.; Zhang, W.Y.; Shield, J.E.; Sellmyer, D.J. Cooperative and noncooperative magnetization reversal in alnicos. AIP Advances 2017, 7, doi:10.1063/1.4976216.
  3. Shield, J.E.; Zhou, J.; Aich, S.; Ravindran, V.K.; Skomski, R.; Sellmyer, D.J. Magnetic reversal in three-dimensional exchange-spring permanent magnets. Journal of Applied Physics 2006, 99, doi:10.1063/1.2163837.
  4. Wu, D.; Yue, M.; Liu, W.Q.; Chen, J.W.; Yi, X.F. Magnetic domain switching in Nd-Fe-B sintered magnets with superior magnetic properties. Materials Research Letters 2018, 6, 255-260, doi:10.1080/21663831.2018.1437571.
  5. Nishino, M.; Uysal, I.E.; Hinokihara, T.; Miyashita, S. Dynamical aspects of magnetization reversal in the neodymium permanent magnet by a stochastic Landau-Lifshitz-Gilbert simulation at finite temperature: Real-time dynamics and quantitative estimation of coercive force. Physical Review B 2020, 102, doi:10.1103/PhysRevB.102.020413.

Comment #2

The Simulation Results part of the paper describes the measurement, but it is not clear, that the given results are how new for the results of other papers.

Response #2

To clearly show the results of this works, the conclusions part is rewritten.

Conclusions

In summary, we have explored the magnetization and the demagnetizing field via micromagnetic simulations to clarify the energy product in multidomain reversal mode. Especially, the energy product calculated by the product between the mean value of B-field and H-field was compared with the energy product calculated faithfully to their essence in consideration of the three-dimensional components of magnetization and demagnetizing field. The simulation results show the obvious degradation of the energy product com-paring the one-dimensional energy product calculated by integration, because the distributions of the magnetizations and the demagnetizing fields are different in multidomain reversal mode. However, the energy product considering three-dimensional components is 28% larger than the energy product calculated from the conventional way. To obtain correct energy product, therefore, magnetic alignment, demagnetizing factor determined by shape of the magnet, and reversal mode should be considered.

Reviewer 3 Report

The authors calculate the energy product of a multi-grain permanent magnet by calculating the BH product for each grain and then doing an average to obtain the BH of the whole magnet. It is a curious result since the usual way of calculating is to use the maximum second quadrant value of B for the entire magnet multiplied by the corresponding value of H. This quick and dirty method is used in the laboratory for a fast evaluation of magnets. Surprisingly enough the fast method is not too far from the more exact method used here.

The authors considered an isotropic magnet in their work. It would be interesting to see if their result is also reasonable for an oriented magnet. 

Real magnets also have intergranular phases present. How would the present results take into account the intergranular phases and what changes are expected?

Author Response

Response to reviewer #3:

Response

We are glad to receive your comprehending statement about our manuscript, and we submitted additional results of the simulation, which of the oriented magnet.

Comment #1

The authors considered an isotropic magnet in their work. It would be interesting to see if their result is also reasonable for an oriented magnet.

Response #1

The model system of Nd2Fe14B with the same grain structure was calculated. The distribution of the easy axes of grains follows the tanθ-type Gaussian distribution with the value of σ = 0.55, which relation is expressed below.

Grain alignment distribution curve is plotted in Figure 1, and the distribution of the alignment results in α = Br/Js = 0.9 [1,2]. From the simulation results shown in Figure 2, the energy products calculated by integration are almost same because the magnetization have the directionality. On the other hand, the value of the energy product from conventional way is larger than the practical values. The results mean that there is still exaggeration by multiplying the average values of B and H-field in the case of the aligned magnet.

Figure 1. Grain alignment distribution curve with σ = 0.55.

Figure 2. Energy product BH calculated by three different ways.

Comment #2

Real magnets also have intergranular phases present. How would the present results take into account the intergranular phases and what changes are expected?

Response #2

As the referee suggested, most of real magnets have intergranular phase inside the magnet, and the internal properties of the magnet should be considered to obtain correct energy product. However, measurement of the internal properties is still challenging. Thus, the energy product by the microstructure should be more investigated to estimate correct energy product of the magnet, because the integrated energy product is largely influenced by its microstructure.

Reviewer 4 Report

In this paper, the authors proposed that the calculated energy product by the integration method is 20% lower than the energy product obtained by the conventional way. Discrepancy between the distribution of the magnetizations and the demagnetizing fields at the reversal process results in the decrease of the energy product. However, this manuscript has some shortcomings and should be further revised before it is considered for publication. Let us elaborate on some of them.

1The clarity of figures 2 is too low. It is recommended to modify and improve the clarity of the picture. Figure 1 is not centered. It is recommended to unify the font at the bottom of the picture.

2The fifth paragraph of the introduction (below picture 2) "combined together." There should be no indentation.

3There is a big problem in the format of the article. The picture should be placed after the complete paragraph rather than inserted into the paragraph. The position of pictures 4 and 5 is unreasonable, so it is suggested to modify.

4 I think the abstract should be rewritten and highlights the novelty.

5The format of the article, especially the recently 3 years reference.

6I think the graphic abstract should be improved.

7 The introduction should be improved and tell the differences compared with others.

8The font of the acknowledged segment is inconsistent. It is recommended to modify it.

9The references cited in the article should be relevant to the content of the article. Please quote reasonably. At the same time, the author is requested to standardize the format of references. Cited the model and artificial intelligence methods in energy element and state of health such as DOI: 10.1155/2021/6693690; 10.1155/2022/9645892; 10.1016/j.energy.2022.123773. The life prediction is more and more important in the energy storage devices.

In short, in its current form, the paper is not suitable for acceptance. The paper needs further modification, by addressing the above-mentioned comments.

Author Response

Response to reviewer #4:

Overall Comments

In this paper, the authors proposed that the calculated energy product by the integration method is 20% lower than the energy product obtained by the conventional way. Discrepancy between the distribution of the magnetizations and the demagnetizing fields at the reversal process results in the decrease of the energy product. However, this manuscript has some shortcomings and should be further revised before it is considered for publication. Let us elaborate on some of them.

Response

We are thankful for the referee’s meticulous comments about our manuscript, and we were pleased to revise it with your suggestion.

Comment #1

The clarity of figures 2 is too low. It is recommended to modify and improve the clarity of the picture. Figure 1 is not centered. It is recommended to unify the font at the bottom of the picture.

Response #1

As the referee suggested, all figures are changed with the figures having higher resolution to improve the clarity, and the original files of figures will be also submitted. In addition, figure 1 and 5 are centered. We checked the font of all contents including pictures and captions, and unified them.

Comment #2

The fifth paragraph of the introduction (below picture 2) "combined together." There should be no indentation.

Response #2

The indentation is removed.

Comment #3

There is a big problem in the format of the article. The picture should be placed after the complete paragraph rather than inserted into the paragraph. The position of pictures 4 and 5 is unreasonable, so it is suggested to modify.

Response #3

As the referee suggested, the position of figure 4 and 5 moved to the end of the paragraph.

Comment #4

I think the abstract should be rewritten and highlights the novelty.

Response #4

The abstract was not clear to suggest the novelty of our works. Thus, we enhanced the additional explanation for the model system, and the critical error of the data was revised.

Abstract

Conventional energy product calculated by product of B-field and H-field is not sufficient for representing the performance of a magnet, because it considers the homogeneous and only uniaxial magnetic properties of the magnet. The energy product has been compared with an energy product by integrating the scalar product of B-field and H-field of each cell with the consideration of three-dimensional components. We investigated a model system by micromagnetic simulation using finite differential method (FDM), and calculated full hysteresis of the magnet. The model system of a Nd2Fe14B magnet composed of grains with about 100 nm in diameter was assumed. In the case of the isotropic multi-grain magnet, the calculated energy product by the three-dimensional integration method is 28% larger than the energy product obtained by the conventional way, though the discrepancy between the distribution of the magnetizations and the demagnetizing fields at the reversal process results in the decrease of the energy product.

Comment #5

The format of the article, especially the recently 3 years reference.

Response #5

We are sorry for missing the difference in the format of the article. The format of the article is checked one-by-one, and is revised.

  1. Kim, N.; Han, H.-S.; Lee, S.; Kim, M.-J.; Jung, D.-H.; Kang, M.; Ok, H.; Son, Y.; Lee, S.; Lee, K.-S. Geometric effects in cylindrical core/shell hard–soft exchange-coupled magnetic nanostructures. Journal of Magnetism and Magnetic Materials 2021, 523, 167599. https://doi.org/10.1016/j.jmmm.2020.167599.
  2. Raviolo, S.; Arciniegas Jaimes, D.M.; Bajales, N.; Escrig, J. Wave reversal mode: A new magnetization reversal mechanism in magnetic nanotubes. Journal of Magnetism and Magnetic Materials 2020, 497, 165944. https://doi.org/10.1016/j.jmmm.2019.165944.

Comment #6

I think the graphic abstract should be improved.

Response #6

As the referee suggested, the graphical abstract is changed to show the difference between the energy product from the conventional method and the energy product calculated by three-dimensional integration.

Comment #7

The introduction should be improved and tell the differences compared with others.

Response #7

As the referee suggested, the introduction part has been rewritten to clarify the novelty of our work with relevant references. Please see the 3-5 paragraph in our revised manuscript and see Refs. [11-14].

The difference comes from distribution of the magnetizations and the demagnetizing fields in the magnet. Thus, the influence by internal magnetic state during magnetic switching is inevitable for the energy product. The switching process is already investigated by many researchers, [7-10] however, its effect on energy product of the permanent magnets has not been studied.

Therefore, in this paper, we investigated integrated energy product from magnetization and demagnetizing fields of cells to obtain the correct energy product, and compared it to the energy product calculated by the conventional method.

Firstly, the conventional energy product assumes the magnetization and the demagnetizing field as uniaxial components in spite of the three-dimensional emission of the magnetic field. Another difference comes from the discrepancy between the distribution of the magnetizations and the demagnetizing fields in the magnet. Because the permanent magnets are not used in saturation state, the influence by the internal magnetic state during magnetic switching is inevitable for determining the energy product of the magnet. Most of studies for switching process cover the dynamic reversal process of the soft mag-nets [[7-12], and there are some researches for investigating reversal mechanism of permanent magnets [13-15]. The reversal process is studied for estimating the effect of temperature [15], exchange-spring [13], and grain boundary diffusion [14].

However, the effect of the reversal mode on the energy product has scarcely been studied despite its importance in estimating the practical magnetic energy of the permanent magnets. At first, the reason why the lack of studies for estimating the energy product based on the theory is the difficulty in measuring the internal state of the magnet. Last but not least, the easy way to estimate the energy product, which multiplies average values of B and H, is widely used. Recently, with the development of the computing technology, studies for magnetic proper-ties inside the magnet by computational methods became further easier.

Therefore, in this paper, we investigated three-dimensionally integrated energy product from magnetization and demagnetizing fields of cells to obtain the correct energy product, and compared it to the energy product calculated by the conventional method.

Additional References

  1. Leighton, B.; Pereira, A.; Escrig, J. Reversal modes in asymmetric Ni nanowires. Journal of Magnetism and Magnetic Materials 2012, 324, 3829-3833, doi:10.1016/j.jmmm.2012.06.023.
  2. Skomski, R.; Ke, L.; Kramer, M.J.; Anderson, I.E.; Wang, C.Z.; Zhang, W.Y.; Shield, J.E.; Sellmyer, D.J. Cooperative and noncooperative magnetization reversal in alnicos. AIP Advances 2017, 7, doi:10.1063/1.4976216.
  3. Shield, J.E.; Zhou, J.; Aich, S.; Ravindran, V.K.; Skomski, R.; Sellmyer, D.J. Magnetic reversal in three-dimensional exchange-spring permanent magnets. Journal of Applied Physics 2006, 99, doi:10.1063/1.2163837.
  4. Wu, D.; Yue, M.; Liu, W.Q.; Chen, J.W.; Yi, X.F. Magnetic domain switching in Nd-Fe-B sintered magnets with superior magnetic properties. Materials Research Letters 2018, 6, 255-260, doi:10.1080/21663831.2018.1437571.
  5. Nishino, M.; Uysal, I.E.; Hinokihara, T.; Miyashita, S. Dynamical aspects of magnetization reversal in the neodymium permanent magnet by a stochastic Landau-Lifshitz-Gilbert simulation at finite temperature: Real-time dynamics and quantitative estimation of coercive force. Physical Review B 2020, 102, doi:10.1103/PhysRevB.102.020413.

Comment #8

The font of the acknowledged segment is inconsistent. It is recommended to modify it.

Response #8

The font of the whole contents including the acknowledged segment was checked and unified.

Comment #9

The references cited in the article should be relevant to the content of the article. Please quote reasonably. At the same time, the author is requested to standardize the format of references. Cited the model and artificial intelligence methods in energy element and state of health such as DOI: 10.1155/2021/6693690; 10.1155/2022/9645892; 10.1016/j.energy.2022.123773. The life prediction is more and more important in the energy storage devices.

Response #9

The references are repeatedly checked, and is standardized for MDPI format including DOI. We cannot understand the relation between our work and cited papers.

Round 2

Reviewer 1 Report

Thank you for completing the work, the manuscript may be published in present form.

Author Response

Response to reviewer #1:

Overall Comments

Thank you for completing the work, the manuscript may be published in present form.

Response

We appreciate the positive statement for our work by the referee.

Reviewer 2 Report

Dear Authors,

 Thank you for your work on the paper. There is a new section from state of the art before the description of the novelty now, it's much better to understand it. However, the methodology part of the paper only describes the examined magnet and mentions the mumax3 software, but please describe the measurement process, but

- what is the difference between this methodology and standard software usage? 

- How is your results can be repeatable from this information?

- which kind of other tools/equations were used for the simulation/creating the physical model please share some more scientific details from the measurement process.

In Simulation Results section, the Fig 4 and Fig 5 contains important results which should be discussed better in the paper.

Author Response

Response to reviewer #2:

Overall Comments

Thank you for your work on the paper. There is a new section from state of the art before the description of the novelty now, it's much better to understand it. However, the methodology part of the paper only describes the examined magnet and mentions the mumax3 software, but please describe the measurement process, but

- what is the difference between this methodology and standard software usage? 

- How is your results can be repeatable from this information?

- which kind of other tools/equations were used for the simulation/creating the physical model please share some more scientific details from the measurement process.

In Simulation Results section, the Fig 4 and Fig 5 contains important results which should be discussed better in the paper.

Response

We are thankful for the additional statement about our manuscript, and we were pleased to revise it with your suggestion.

Comment #1

What is the difference between this methodology and standard software usage?

Response #1

There are several ways for obtaining the energy product of a magnet from the experimental results. But, let’s take the BH tracer as a representative example. When a sample is placed coaxially in a periodically varying magnetic field, the magnetization in the sample also changes periodically. This variation can be picked up by a search coil which is placed coaxially with the sample, and the schematic images of the BH tracer are presented in Figure 1. By following equation, the search coil gives the value of J, which results in BH values.

Figure 1. The schematic images of the BH tracer.

However, this measurement has several assumptions. First of all, the stray fields from the magnet in the empty space between the search coil and the sample (Ac – As) are ignored, which have small intensity, but still not zero. Another point is the uniaxial consideration of the measurement (Magnet emits three-dimensional magnetic field). Last but not least, it regards the magnetization and the demagnetizing field of the sample as a value, which makes the difference with the practical energy product of the sample. In conclusion, standard software regards the magnetization and the demagnetizing field as a value, but the real value of the magnetization and the demagnetizing field of the sample is composed of many of different values along three-dimensional direction.

Comment #2

How are your results can be repeatable from this information?

Response #2

Our works can represent the tendency of the energy product about the isotropic permanent magnet, however, it cannot cover the all kinds of reversal modes in magnets. Because measuring specific components of the magnetization and the demagnetizing field in the magnet is still challenging, the energy product of many models for various magnet should be studied in detail, and the energy product should be estimated by the right models.

Comment #3

which kind of other tools/equations were used for the simulation/creating the physical model please share some more scientific details from the measurement process.

Response #3

We are sorry for missing the fundamental information about the simulation. The contents about the energy considered in the simulation are additionally reinforced. The scientific details of the simulation are given in the paper, the rest is a computational technique by using Matlab code, which cannot be opened.

The hysteresis loop was computed by minimization of the Gibbs free energy for external magnetic field ranging from – 10 to + 10 T, which was applied along the cartesian z-axis. The total magnetic Gibbs free energy which is described by the below equation.

E = Eex + EK + Ed + EZeeman,

(2)

where Eex, EK, Ed, and EZeeman are the exchange energy, the anisotropy energy, the magnetostatic energy, and the Zeeman energy, respectively.

Comment #4

In Simulation Results section, the Fig 4 and Fig 5 contains important results which should be discussed better in the paper.

Response #4

As the referee suggested, we added more discussion about Fig. 4 and Fig. 5 in Simulation Results section.

Demagnetizing factor (N = –Hd/Mz) is geometry-dependent value representing the magnet, which means the proportionality of the demagnetizing field with the magnetization [26]. To clearly show the mismatch between the magnetization and the demagnetizing field, the demagnetizing factor of each cell was investigated, and a histogram for the distribution of the demagnetizing factor is given in Figure 5. At the saturation state, the demagnetizing factor is affected by the distribution of the demagnetizing field and follows a narrow Gaussian distribution because the magnetizations in the magnet are aligned along the applied field direction. On the other hands, in the case of the remanent state, the demagnetizing factor spreads over a wide range due to the mismatch between the magnetizations and the demagnetizing fields.

For these difference of the distribution between the magnetization and the demagnetizing field during the magnetic reversal, the energy product calculated from the integration of each element is inevitably different with the energy product given by the product of mean B and H.

Round 3

Reviewer 2 Report

The authors answered all of my questions